# Blue Light Enhances Health-Promoting Sulforaphane Accumulation in Broccoli (*Brassica oleracea* var. *italica*) Sprouts through Inhibiting Salicylic Acid Synthesis

**DOI:** 10.3390/plants12173151

**Published:** 2023-09-01

**Authors:** Youyou Guo, Chunyan Gong, Beier Cao, Tiantian Di, Xinxin Xu, Jingran Dong, Keying Zhao, Kai Gao, Nana Su

**Affiliations:** College of Life Sciences, Nanjing Agricultural University, Nanjing 210095, China; 2022116037@stu.njau.edu.cn (Y.G.); 2015116010@stu.njau.edu.cn (C.G.); 2022116002@stu.njau.edu.cn (B.C.); 2022816098@stu.njau.edu.cn (T.D.); 9211010221@stu.njau.edu.cn (X.X.); xyz15380819376@163.com (J.D.); 9211010219@stu.njau.edu.cn (K.Z.); 18222826021@163.com (K.G.)

**Keywords:** broccoli sprouts, salicylic acid, sulforaphane, blue light, gene expression, anti-cancer

## Abstract

As a vegetable with high nutritional value, broccoli (*Brassica oleracea* var. *italica*) is rich in vitamins, antioxidants and anti-cancer compounds. Glucosinolates (GLs) are one of the important functional components widely found in cruciferous vegetables, and their hydrolysate sulforaphane (SFN) plays a key function in the anti-cancer process. Herein, we revealed that blue light significantly induced the SFN content in broccoli sprouts, and salicylic acid (SA) was involved in this process. We investigated the molecular mechanisms of SFN accumulation with blue light treatment in broccoli sprouts and the relationship between SFN and SA. The results showed that the SFN accumulation in broccoli sprouts was significantly increased under blue light illumination, and the expression of SFN synthesis-related genes was particularly up-regulated by SA under blue light. Moreover, blue light considerably decreased the SA content compared with white light, and this decrease was more suppressed by paclobutrazol (Pac, an inhibitor of SA synthesis). In addition, the transcript level of SFN synthesis-related genes and the activity of myrosinase (MYR) paralleled the trend of SFN accumulation under blue light treatment. Overall, we concluded that SA participates in the SFN accumulation in broccoli sprouts under blue light.

## 1. Introduction

Sulforaphane (SFN) is an important secondary metabolite in cruciferous vegetables, which is hydrolyzed from glucosinolates (GLs). Moreover, it is identified as the most active health-promoting compound among analogs produced by the GLs [1]. However, GL itself has no significant activity, so it needs to break the glucoside bond between sulfur atoms and glucose through myrosinase (MYR) to form unstable aglucone. The unstable aglucone is then rearranged to form isothiocyanate substances that are beneficial to the human body, of which the most representative substance is SFN [2]. At present, a large number of studies have shown that SFN has a detoxification effect on carcinogens and can inhibit the growth of cancer cells [3]. In addition, SFN has anti-oxidation, antibacterial, anti-inflammation and anti-aging effects on the human body. Importantly, SFN is by far recognized as one of the strongest anti-cancer and cancer-preventing natural active ingredients in vegetables [4,5,6]. Therefore, SFN has attracted great attention from scientists as a potential cancer preventive agent, and improving the accumulation of SFN has become a new health hotspot.

At present, the synthetic pathway of SFN has been elucidated. The synthesis of SFN starts with methionine as the substrate, then goes through the extension of amino acids to the formation of aldoxime, then through oxidation, cleavage and sulfation to form GLs, and finally are hydrolyzed to SFN by MYR [7]. In recent years, there have been many research reports on the regulation of SFN synthesis. For example, through major quantitative trait locus (QTL) analysis, Lambrix found that epithiospecifier protein (ESP) can hydrolyze GLs [8]. In *Brassica rapa*, *MYB28* is a key transcription factor for SFN synthesis [9]. In addition, many other genes in the biosynthetic pathway of SFN have been identified, such as *MYR*, *Elong*, *CYP79B2*, *CYP83B1*, *UGT74B1* and so on [10]. In addition, the synthesis of SFN is also regulated by other substances. In the last ten years it has been found that the treatment of exogenous glucose, sulfur and selenium can improve the content of SFN in broccoli [11,12]. Riahi-Madvar and Manal Ali et al. also found that the accumulation of SFN in cruciferous vegetables could be regulated by spraying exogenous substances such as NaCl and plant hormones during the growth period [13,14]. Similar research has also been found in cabbage and radish [15].

Light is a very important environmental signal throughout the life cycle of plants [16]; it also plays an important role in regulating the synthesis of SFN. Studies have shown that LED red light can promote callus proliferation and SFN accumulation in *Raphanus sativus* L. [17]. The short duration of blue light treatment before harvest can significantly increase the accumulation of SFN content in *Brassica oleracea* [18]. Meanwhile, in broccoli sprouts, Kopsell found that different ratios of LED light treatment can improve the SFN content [19]. In addition, the study found that the amount of GLs, as the precursor of SFN, may be affected by different doses of blue light [20]. Zheng et al. found that blue light can effectively improve the total content of GLs in cabbage [21]. Coincidentally, the same result was also reported in the study of GLs in Chinese kale sprouts [22]. On the contrary, there are also some different points of view. In *Cardamine fauriei*, the content of aliphatic GLs can be improved through the use of blue light [23]. However, in *Brassica napus*, the total amount of GL content does not respond to blue light [24]. Because of these conflicting results, the regulation of SFN in plants using blue light needs more verification.

As a common hormone, salicylic acid (SA) plays an indispensable function in secondary metabolism in plants [25]. In recent decades, it has been found that SA, as a signaling molecule, can regulate the formation of secondary metabolites such as GLs in plant physiological processes [26]. The exogenous application of SA at a suitable concentration can achieve the effect of improving the GL composition and content in cruciferous vegetables [27]. Pérez-Balibrea et al. found that indole GLs content increased by 33% under 100 μM SA treatment in 7 d broccoli sprouts [28]. Furthermore, it was reported that increased SFN concentration might be related to SA accumulation in plants [29]. Coincidentally, recent research reported that SA can suppress the accumulation of indole-3-carbinol (I3C, another GLs hydrolysate) in broccoli sprouts [30]. However, there are still few studies on whether SA and blue light can co-regulate the SFN content in broccoli.

At present, the reports on SFN mainly focus on the anti-cancer aspect [31], while there are few studies on its accumulation in plants, especially on the optimal time point and mechanism of SFN accumulation in plants under the treatment of blue light. Furthermore, the current research on GLs is mainly concentrated on *Arabidopsis thaliana*, while there are still few reports on other cruciferous plants, such as brassicas crops [32]. As a common vegetable in the cruciferous family, broccoli sprouts are one of the potentially functional foods. They are favored by consumers because of their rich beneficial substances, such as glucoraphanin. Furthermore, the SFN concentration in broccoli sprouts is 20-fold higher than that in mature broccoli [33]. Therefore, it is very important to explore the effect of different lights to improve the quality of broccoli sprouts. The research was to investigate the function of blue light treatments on SFN content and MYR activity and the induction mechanism was further elucidated by studying the effect of SA on transcription levels of SFN synthesis-related genes in broccoli sprouts.

## 2. Results

### 2.1. Effects of Blue Light on the Growth of Broccoli Sprouts

Light is a key environmental factor regulating plant growth. To explore the effect of blue light on the growth of broccoli sprouts, we analyzed the phenotypes and growth parameters [height and fresh weight (FW)] of broccoli sprouts with blue light and white light treatments, respectively. As shown in Figure 1, the height of broccoli sprouts under blue light treatment gradually increased with time and was always higher than that under white light treatment. Compared with the control (white light treatment), the height of 5/6/7 d broccoli sprouts with blue light treatment was increased by 16.95%, 8.67% and 13.60%, respectively (Figure 1B). In addition, the fresh weight of 5/6 d broccoli sprouts under blue light treatment was significantly higher than that of control, but the weight of 7 d broccoli sprouts showed no significant difference between white light and blue light treatment. Furthermore, the fresh weight of 5/6/7 d broccoli sprouts under blue light treatment was increased by 12.67%, 9.58% and 1.22%, respectively, compared to the control (Figure 1C). Therefore, according to Figure 1, we found that blue light can promote the growth of broccoli sprouts.

### 2.2. Effects of Blue Light on the Content of SFN and MYR Activity

To further verify the effect of blue light on the growth and development of broccoli sprouts, we analyzed sulforaphane (SFN) content and myrosinase (MYR) activity under the treatment of blue light at different times. It was found that the SFN content after 5/6/7 d blue light treatment was 94%, 139% and 90% of that after white light treatment. In addition, after 6 d blue light treatment, SFN content in broccoli sprouts was the highest (Figure 2A). Previous studies have found that the activity of MYR, a key enzyme in SFN production, also showed a similar trend [4,5,6]. The MYR activity of 6 d broccoli sprouts treated with blue light was increased by 19.16%, while it was markedly reduced in 5 d and 7 d broccoli sprouts under blue light treatment in comparison with that of the controls. In addition, MYR activity was also the greatest in 6 d broccoli sprouts under blue light treatment (Figure 2B). Together, these results indicated that blue light significantly increased SFN accumulation and MYR activity in 6 d broccoli sprouts. Moreover, the accumulation of SFN and the activity of MYR exhibited the same trend under the treatment of blue light, which was consistent with the previous results.

### 2.3. Effect of Blue Light on Antioxidant Enzyme Activity of Broccoli Sprouts

Different light treatments can change the activity of protective enzymes in plants, thereby inducing different degrees of removal mechanisms of reactive oxygen species in plants (such as SOD, POD and CAT), thus affecting plant growth and secondary metabolism [34]. Therefore, in order to verify whether blue light has an effect on the activity of protective enzymes in broccoli sprouts, we determined the activity of reactive oxygen species (ROS) in broccoli sprouts treated with blue light at different times, including SOD, POD and CAT. The activity of SOD, POD and CAT of 6 d broccoli sprouts revealed that blue light treatment did not change SOD activity but decreased the activity of POD and CAT as compared with that of the control (Figure 3). At the same time, by contrast, the POD and CAT activities of 6 d broccoli sprouts were obviously decreased by 51.12% and 36.8%, respectively, under blue light treatment. The activities of POD and CAT also showed similar phenomena in 7 d broccoli sprouts. By contrast, POD and CAT activities of 5 d broccoli sprouts were dramatically enhanced under blue light (Figure 3B,C). Interestingly, there was no significant difference in SOD activity between the control and blue light treatment (Figure 3A).

### 2.4. Effects of Blue Light on the Expression Level of Related Genes of SFN Synthesis

According to the above results, we found that blue light can promote the accumulation of SFN. To elucidate the molecular mechanism, we determined the expression level of SFN biosynthesis-related genes in broccoli sprouts. *Elong*, *CYP83A1*, *UGT74B1*, *ST5b*, *MYR* and *ESP* were key genes in the SFN biosynthesis pathway of brassica species [35,36,37]. All these genes except *BoESP* in 6 d broccoli sprouts were up-regulated with blue light treatment. As shown in Figure 4, the expression of *BoElong*, *BoCYP83A1*, *BoUGT74B1*, *BoST5b* and *BoMYR* in 6 d broccoli sprouts under blue light treatment was increased by 427%, 1201%, 1368%, 429% and 453%, respectively, compared with the control. However, under blue light treatment, their expression level in 5 d broccoli sprouts was decreased by 56.4%, 70%, 14.6%, 34% and 82.6%, respectively, as compared with that of the control (Figure 4A–E). Similar results were also found in 7 d broccoli sprouts under blue light. Meanwhile, there was no effective difference from the control of *BoESP* expression under blue and white light treatments (Figure 4F).

Based on gene heat map analysis under different conditions, we found that the transcription level of these genes under blue light for 6 d was significantly higher than others, except for *BoESP* (Figure 5). Therefore, in this research, the 6 d broccoli sprouts would be used for further experiments.

### 2.5. Effects of Blue Light on Endogenous Hormone Content, Expression of Key Genes and Enzymatic Activity of SA Synthesis Pathway

Phytohormones play a vital role in the growth, development and secondary metabolism of plants [25]. To explore whether light affects the production of phytohormones, which in turn affects the metabolism of plant secondary substances, we tested the content of a variety of basic phytohormones in broccoli sprouts under blue light. The results showed that the SA content was significantly reduced by 24.48% compared with the control after the 6 d broccoli sprouts were irradiated by blue light (Figure 6A). However, there were no significant differences in auxin, ethylene (ETH), gibberellin (GA), cytokinin (CTK), and abscisic acid (ABA) contents, which were all compared with white light (Figure 6A). Compared with other hormones, SA decreased most significantly under blue light. Therefore, the next step is to study the mechanism of the effect of blue light on SA metabolism.

In order to investigate the mechanism of blue light on SA metabolism, we examined the activities of key enzymes in SA synthesis, namely PAL and BA2H [30]. As shown in Figure 6, consistent with the changes in SA content after blue light treatment, the transcription level of the SA synthesis-related genes *BoPAL*, *BoBA2H* and the corresponding enzymatic activities were decreased in broccoli sprouts (Figure 6B,C). Meanwhile, the expression level of *BoPAL* and *BoBA2H* in broccoli sprouts treated with blue light was decreased by 82.2% and 96.4%, respectively, compared with those treated with white light (Figure 6B). As a similar trend in enzymatic activities, the activities of PAL and BA2H were significantly inhibited after blue light treatment and decreased by 82.2% and 96.4%, respectively, compared with control (Figure 6C). Together, the result indicated that blue light can inhibit SA metabolism in 6 d broccoli sprouts. However, 6 d broccoli sprouts treated with blue light significantly accumulated SFN. Therefore, we speculated that SA may inhibit the accumulation of SFN, and there is a negative correlation between SA and SFN.

### 2.6. Effects of Exogenous SA on Content of Endogenous SA, SFN Synthesis

To verify our hypothesis, we investigated the effects of exogenous SA on endogenous SA content and SFN synthesis. As shown in Figure 7A, under white and blue light treatments, compared with control, the SA content of broccoli sprouts treated with exogenous SA was increased by 12.5% and 12.4%, respectively. The content of SA was decreased significantly by 11.1% and 11.2%, respectively, with Pac (an SA synthesis inhibitor) treatment. However, co-treatment of SA plus Pac obviously improved the content of SA compared with that of Pac treatment alone (Figure 7A).

By contrast, under white and blue light treatment, the content of SFN is completely opposite to that of SA. Under SA treatment, the SFN content of white and blue light was significantly reduced by 16.1% and 12.1%, respectively, in comparison with that of the controls. The broccoli sprouts with Pac treatment resulted in rapid recovery of SFN, even higher than that of broccoli sprouts treated with blue light alone. When SA was co-treated with Pac, the inhibition of SFN was significantly enhanced (Figure 7B). There was a similar tendency in MYR activity in broccoli sprouts. MYR activity of broccoli sprouts was reduced by SA treatment, while it was significantly increased by Pac treatment. The decrease in MYR activity was effectively alleviated after co-treatment of Pac and SA compared to SA alone (Figure 7C). These results correspond exactly to our hypothesis, which indicates that SA can inhibit the synthesis of SFN.

### 2.7. Effects of SA on the Expression of SFN Synthesis Genes

Based on the above results, in order to further verify the molecular mechanism of SA inhibiting SFN, we measured the transcription levels of SFN synthesis-related genes in broccoli sprouts treated with SA, Pac, and co-treatment of SA and Pac. As shown in Figure 8, compared with white light treatment, blue light enhanced the expression of SFN synthesis related genes, except for *BoESP* (Figure 8), while, under blue light, SA treatment reduced their expression levels. In addition, in the absence of SA, the expression levels of *BoElong*, *BoCYP83A1*, *BoUGT74B1*, *BoST5b*, and *BoMYR* increased by 40.3%, 21.4%, 14.6%, 17.9% and 27.9%, respectively, in Pac treatment alone compared with SA treatment alone. However, co-treatment of SA and Pac had no obvious effect on their expression in comparison with the SA treatment (Figure 8A–E). At the same time, the transcription levels of these genes under white light were similar to those under blue light. No matter whether under white or blue light treatment, the expression of these genes was remarkably decreased by SA. However, the expression level of *BoESP* was not significantly changed by SA under white or blue light treatment (Figure 8F). In summary, SA participated in the regulation of SFN accumulation in broccoli sprouts under blue light; that is, blue light inhibited the negative effects of SA on SFN and further promoted the accumulation of SFN.

### 2.8. Verification of SA Biosynthesis-Deficient Arabidopsis Mutants

SFN is an important secondary metabolite in plants, which mainly exists in *Brassicaceae plants*. *Brassicaceae* plants *Arabidopsis thaliana*, as a model plant, can also generate SFN. To further demonstrate the negative effect of SA on SFN accumulation, we used SA biosynthesis-deficient Arabidopsis mutants for further analysis. The laboratory has carried out some work on the mutant, including purification and verification, gene expression of *eds5* and *ics1*-L1, SA content, and found that blue light inhibits the synthesis of SA. Furthermore, we also found that the seedlings of *eds5* and *ics1*-L1 displayed shorter hypocotyl and root compared to the wild-type (WT) [30].

Next, we detected the SFN content of *eds5* and *ics1*-L1 after two weeks of treatment with blue light and white light, respectively. The results of Figure 9 showed that the SFN content of WT, *eds5 ics1*-L1 under blue light treatment was higher than that of white light treatment. Further, the SFN content of *eds5* and *ics1*-L1 of SA biosynthesis-deficient Arabidopsis mutants was significantly higher than that of WT. According to the results, we further demonstrated that blue light promotes SFN accumulation by inhibiting the synthesis of SA.

## 3. Materials and Methods

### 3.1. Plant Growth and Treatments

Broccoli (*Brassica oleracea* var. *italica*, Mantuolv) seeds were bought from the Nanjing Jinshengda Seed Company (Nanjing, China). The seeds were soaked in water at room temperature for 4–6 h and then spread on wet gauze to germinate for 24 h and covered with plastic wrap to retain moisture. At the same time, they were cultured under dark conditions of 25 ± 3 °C until white buds appeared. Then, the buds with consistent growth were selected to cultivate in a seedling tray containing 1/4 Hoagland’s solution [38] and illuminated using a LED with white and blue light in a growth chamber (Ningbo Sai Fu Instrument Co, Ltd., Ningbo, China). The wavelengths of white light and blue light are 380–750 nm and 460 nm, respectively. The photoperiod was set at 16 h light/8 h dark with a light intensity of 200 μmol m^−2^ s^−1^. In addition, the temperature and relative humidity were set at 25 ± 1 °C and 75 ± 5%, respectively. Then, 5/6/7 d broccoli sprouts were collected for experimental analysis.

The *A*. *thaliana eds5* (SALK-091541) and *ics1*-L1 (SALK-133146) mutants were bought from the AraShare. The *A*. *thaliana* was cultivated using the method of Wu et al. with some revisions [39]. The seeds of *Arabidopsis thaliana* were first cultured on 1/2 MS solid medium for about 10 d, and then the seedlings were transferred to the soil and cultivated for 2 weeks under blue light and white light in a growth chamber (Ningbo Sai Fu Instrument Co., Ltd., Ningbo, China) with a light intensity of 200 μmol m^−2^ s^−1^ and a photoperiod of 16 h light/8 h dark. The temperature and humidity settings of the growth chamber are the same as those of broccoli. Two weeks later, the plants were collected for measurement.

### 3.2. SA Treatments of Broccoli Sprouts

In the SA experiment, the 4 d broccoli sprouts were treated with SA using the method of Wang et al. [30]. The SA was dissolved in 0.2% ethanol at 50 μM concentration, and paclobutrazol (Pac, SA synthesis inhibitor) was dissolved in ddH_2_O at 100 μM concentration. The treatments of broccoli sprouts were grouped as follows: (1) Control treatment (distilled water); (2) SA treatment (50 μM); (3) Pac treatment (100 μM); and (4) Pac + SA treatment (100 μM Pac + 50 μM SA). Then, 6 d broccoli sprouts were collected for analysis.

### 3.3. Determination of Growth Parameters

The growth parameters of 5/6/7 d broccoli sprouts (phenotype, sprout height and fresh weight) were determined. The plant height of broccoli sprouts was measured using a ruler (Accuracy: 0.1 cm). The fresh weight of broccoli samples under different treatments was determined using an electronic scale (Accuracy 0.0001 g). The phenotype of broccoli sprouts was captured by the camera (Canon, Toyko, Japan, EOS 6D MarkII).

### 3.4. Determination of SFN Content

SFN was performed using the method outlined by Yang et al. with some revisions [11]. A total weight of 1.5 g fresh sample (Broccoli sprouts and Arabidopsis plants) was added to 12 mL of distilled water to grind it into a homogenate. Then, 10 mL of dichloromethane was added after hydrolyzation at 37 °C for 3 h and then evaporated at 37 °C. After this, the extract was dissolved via ultrasonic means with 10% acetonitrile. Then, the crude extract was passed through a 0.45 μm syringe filter. Finally, the HPLC system was used to separate the extraction liquid with a C18 (ACQUITY UPLC, 2.1 × 100 mm, 1.8 Micron) column.

### 3.5. Enzymatic Activities Assay

Myrosinase (MYR) activity was detected using the method of Guo et al. with some revisions [40]. Fresh broccoli sprouts (1.5 g) were ground in order to homogenize them with 9 mL of 0.1 M pre-cooled phosphate buffer in an ice bath and centrifuged at 12,500× *g* for 0.5 h at 4 °C. Then, 0.5 mL of supernatant was mixed with allyl glucoside solution and reacted for 20 min at 37 °C. The glucose content was measured with a glucose kit (Nanjing Jiancheng Bioengineering Institute, Nanjing, China). In this experiment, 1 nM glucose generated per minute was considered 1 enzyme activity unit (U/g FW).

Phenylalanine ammonia lyase (PAL) activity was analyzed using the method of Lister et al. with minor revisions [41]. Fresh broccoli sprouts (0.5 g) were ground in an ice bath with 5 mL of the extract solution (0.018 M 2-Mercaptoethanol, 0.05 M Vitamin C and 0.1 M pH 8.7 Boric acid buffer). Each homogenate was centrifuged at 14,000× *g* for 15 min at 4 °C. In this experiment, 250 μL of supernatant was mixed with 875 µL Boric acid buffer and 250 µL of 11 mM L-phenylalanine in a new EP tube. The mixture was set in a water bath at 30 °C for 0.5 h, and 0.5 mL of 15% HCl was used to stop the reaction. The product was centrifuged at 8000× *g* at room temperature for 10 min. The absorbance of samples was measured at a 290 nm wavelength using a Spectrophotometer (PGENERAL, Beijing, China). Then, the PAL enzyme activity was calculated via a standard curve.

Benzoate hydroxylase (BA2H) activity was analyzed using the method of León et al. with some revisions [42]. Liquid nitrogen was used to grind 1.5 g fresh broccoli sprouts. The suspension composed of powder and 6 mL of the extraction buffer (1 mM phenylmethylsulphonyl fluoride, 14 mM 2-mercaptoethanol and 20 mM triethanolamine, pH 7.4) was centrifuged at 10,000× *g* for 30 min at 4 °C; then, 0.2 mL of the supernatant was incubated with the reaction mixture (1 μM NADPH, 1 μM BA, 10 μM triethanolamine buffer, pH 7.4) at 30 °C for 0.5 h. Then, the reaction was stopped using 0.25 mL of 15% (*w*/*v*) TCA. After the reaction mixture was eddied and centrifuged for 5 min at 12,500× *g*, 0.5 mL of cyclopentane/ethyl acetate/2-propanol (50/50/1, *v*/*v*/*v*) was applied to leach the supernatant. The upper was evaporated at 35 °C in a vacuum dryer (Hualida, LNG-T1OO, Changzhou, Jiangsu, China), and then the residue was redissolved in 23% (*v*/*v*) methanol/sodium acetate buffer (20 mM, pH 5.0). The SA content was detected via HPLC with an Agilent C18 column (250 × 4.6 mm, 5 μm) at a 300 nm detection wavelength. Methanol–water–ice–acetic acid (48:52:3) was chosen as the mobile phase, and the flow rate is 1.0 mL/min.

Superoxide dismutase (SOD) activity: The extraction of the antioxidant enzymes was slightly modified with reference to the method of Sudhakar et al. [43]. Take 1.5 g broccoli sprouts, homogenate with 4 mL of pre-chilled 50 mM phosphate buffer (pH 7.0, containing 1% polyvinylpyrrolidone and 0.1 mM EDTA) on ice, and centrifuge at 10,000× *g* at 4 °C for 20 min. The clear solution is the crude enzyme solution. The enzyme activity unit (U) is the SOD amount corresponding to an SOD inhibition rate of 50% in 1 mL of reaction solution per gram of fresh broccoli sprouts.

Catalase (CAT) activity was detected by Singh et al. with minor modifications [44]. In this experiment, 1.5 mL of 20 mM H_2_O_2_ was taken and reacted with 50 μL of crude enzyme extract, recorded at 470 nm OD every 60 s for 2 min. The change in absorbance per minute of broccoli sprouts decreased by 0.01 to 1 catalase activity unit (U).

Peroxidase (POD) activity was detected according to the method outlined by Eichholz et al. with slight modifications [44]. Enzyme extract (200 μL) was reacted with 5.8 mL 0.05 M phosphate buffer, 2 mL 2% H_2_O_2_ solution and 2 mL 0.05 M guaiacol. The enzyme solution after boiling for 5 min was used as a control and recorded at 470 nm every 60 s for 2 min. The change of OD 470 nm per minute with fresh broccoli sprouts was 0.01 as 1 enzyme unit (U).

### 3.6. Analysis of Plant Hormone Content

The content of plant hormones SA, IAA, GA, CTK, ABA and ETH were analyzed using the method outlined by Zhang et al. with minor revisions [45]. The sample of 6 d broccoli sprouts with blue light and white light treatments (1.0 g) was grounded into powder in the presence of liquid nitrogen and transferred to a centrifuge tube, and then 30 mL of 80% methanol was added to the centrifuge tube. The mixture was centrifuged at 8500× *g* at room temperature for 20 min. Then, the supernatant was transferred to a new tube, and 15 mL of 80% methanol was added for the precipitation of secondary extraction. Finally, the extract was filtered as in 2.4. The HPLC system was used to separate the extraction liquid with a C18 column (The column type is the same as in 2.4). The compounds were detected based on 280 nm. The column oven temperature was set at 30 °C, and 2 μL portions were injected into the column with a flow rate of 1.0 mL·min^−1^. The proportion of mobile phase was 30% acetate buffer (pH 3.22): 70% (methanol).

### 3.7. RNA Preparation and Quantitative Real-Time-PCR Analysis

Quantitative real-time PCR (qRT-PCR) was used to investigate the effects of various treatments on the expression patterns of genes encoding SFN synthesis and SA synthesis. The broccoli sprouts sample (0.5 g) was homogenized using a mortar and pestle in the presence of liquid nitrogen. The total RNA was extracted using trizol reagent according to the instructions of the RNA extraction kit (Invitrogen, Gaithersburg, MD, USA). After that, the RNA integrity was detected via 1% agarose gel electrophoresis. The cDNA was produced by the reagent kit (TOYOBO CO, LTD. Japan). The final product can be used for qRT-PCR experiments. Concerning qRT-PCR analysis, the expression level of the target genes was determined using the system of Wu [39]. The primers were designed according to Wang et al. and are shown in Table 1 [30]. Finally, the 2^−ΔΔCt^ method was used to calculate the transcription level of different genes [46].

### 3.8. Statistical Analysis

All experiments were performed in triplicate biological and technical replicates. SPSS (Chicago, IL, USA) was used to analyze the data. The data were presented as mean ± standard deviation of three independent experiments after a one-way analysis of variance (ANOVA). The data were analyzed via Duncan’s multiple range test (*p* < 0.05).

## 4. Discussion

In recent years, sprouts like broccoli and radish have become popular with consumers because of their rich nutritional value [47]. According to scientific studies, the induced amount of phase II enzymes in broccoli sprouts is 10–100 times higher than that of mature broccoli. Phase II enzymes are a class of key enzymes in the enzyme system of detoxifying carcinogens for detoxifying carcinogenic factors [48]. Therefore, broccoli sprouts have become a popular healthy vegetable due to their natural active ingredient SFN, which is also rich in biological activities such as cardioprotection and anti-inflammatory activities [40]. In this study, we demonstrated that SA is involved in the blue light-induced SFN accumulation in broccoli sprouts. To our knowledge, what the study shows is that blue light could induce SFN accumulation in broccoli sprouts, while most previous reports focused on the improvement of nutritional substances, such as phenolics [40] and flavonoids [49].

Light is a natural environmental signal that plays an indispensable role in the secondary metabolism of plants. Previous research has shown that blue light can improve the nutritional quality of cruciferous vegetables such as broccoli and Chinese kale [18,22]. Furthermore, blue light can also affect the growth and development of plants [30]. In this study, blue light notably increased the broccoli sprout’s height and fresh weight; the broccoli sprout’s height and fresh weight grown under blue light for 6 d was significantly higher than those under white light treatment (Figure 1B,C). The biomass of broccoli sprouts increased, which may be related to the promotion of leaf stretching and chloroplast development by blue light [50]. Photoinhibition will occur if a plant is in an unfavorable light environment for a long time, which will lead to the accumulation of activity of antioxidant enzymes [51]. Our studies have shown that there was no significant difference in SOD activity between the control and blue light in broccoli sprouts, and the activity of CAT and POD after blue light treatment decreased, which may be because the intensity of blue light did not cause severe oxidative damage and was not sufficient to cause a major outbreak of ROS (Figure 3).

As an important secondary metabolite in cruciferous plants, SFN is also regulated by blue light. Previous research has shown that different ratios of LED light treatment can significantly improve the SFN content in *Brassica oleracea* [19]. This study suggested that blue light treatment obviously enhanced SFN content in 6 d broccoli sprouts (Figure 2A). Similar results have been reported that blue light increased GLs content of Chinese kale sprouts [22]. In the past 10 years, MYR has been shown to be a key enzyme in GL hydrolysis [6]. A previous study has shown that light stimulation can also change MYR activity [52]. In this experiment, we also exhibited that blue light can significantly enhance the activity of MYR, which was also consistent with the change in SFN content (Figure 2B). SFN biosynthesis in plants involves the participation of several genes, such as *Elong*, *CYP83A1*, *UGT74B1*, *ST5b*, *MYR*, *ESP*, etc. [36]. Therefore, RT-qPCR was used to detect the transcriptional levels of these genes. In parallel with the accumulation of SFN, the expression levels of the SFN synthesis-related structural genes (*BoElong*, *BoCYP83A1*, *BoUGT74B1*, *BoST5b*, *BoMYR*) were all significantly up-regulated by blue light illumination, which implied that these five genes may take part in the synthesis of SFN (Figure 4 and Figure 5). Coincidentally, in *Arabidopsis*, Huseby et al. found that blue light could increase the transcription levels of GLs synthesis genes [53]. In a word, the results indicated that blue light promoted SFN accumulation in broccoli sprouts by up-regulating genes related to SFN biosynthesis.

Previous research suggested that blue light could induce GL accumulation [22], but whether SA is involved in this process is still unknown. Our study demonstrated that compared with the white light, 6 d illuminated with blue light significantly inhibited the endogenous SA level in broccoli sprouts (Figure 6A), suggesting that blue light induced accumulation of SFN partly contributed to the decrease in SA level. Similarly, Wang’s results showed that blue light can obviously inhibit the content of SA [30]. SA, as an important plant hormone, is recently considered a central regulator in the synthesis of GLs [26,29,54]. To further confirm this, we quantified the expression of *BoPAL* and *BoBA2H* and tested the corresponding enzymatic activities. The transcript level of *BoPAL* and *BoBA2H* was significantly inhibited by blue light, and a similar trend was observed in enzymatic activities (Figure 6B,C), which is consistent with Wang’s research [30]. In conclusion, these results revealed that blue light illumination inhibited SA synthesis, mainly through down-regulating *BoPAL* and *BoBA2H*, finally increasing the accumulation of SFN. On the contrary, other studies have found that light can promote SA accumulation in *Solanum lycopersicum* [55] and *Helianthus annuus* [56]. Because of these opposing results, a great deal of evidence is required for further verification.

In recent years, it has been found that the exogenous application of SA can regulate the nutritional quality and growth of vegetables [27,57]. In this research, according to the analysis of exogenous SA and Pac results, we found that the exogenous application of SA significantly decreased the content of SFN and MYR activity in broccoli sprouts under blue light (Figure 7B,C), which implied that SA can inhibit SFN synthesis in broccoli sprouts. This phenomenon was mainly due to the decreased transcription level of genes related to SA synthesis under blue light in *Brassica oleracea* [30]. Many studies suggested that SA treatment can change the transcription levels of genes related to GLs [26]. This study also found that the transcription level of genes related to SFN synthesis was down-regulated under SA treatment (Figure 8). Moreover, it was found that the content of SFN in SA biosynthesis-deficient Arabidopsis mutants was significantly higher than that in the wild type. The SFN content under the blue light treatment was higher than that under the white light treatment (Figure 9). All these results indicated that SA is involved in blue light-induced SFN accumulation. Contrariwise, in *Arabidopsis*, SA treatment markedly induced the gene expression of *MYB51*, which plays an important role in the synthesis of indole GLs [29,58]. In a word, these data indicate that SA is involved in the regulation of genes related to GL synthesis in cruciferous plants, thus regulating the metabolism of GLs and its hydrolysates.

## 5. Conclusions

In summary, the content of SFN in broccoli sprouts under blue light is significantly increased, and the expression of genes related to SFN synthesis is also significantly increased. The exogenous addition of Pac under blue light further enhanced the content of SFN compounds and the expression of synthetic genes. However, the exogenous addition of SA inhibited the expression of SFN synthesis-related genes significantly. Considering different extents of decrease in SFN synthesis gene expression with SA treatment under blue light, we speculate that SA inhibited SFN accumulation partly because of blue light induction (Figure 10).

## Figures and Tables

**Figure 1 plants-12-03151-f001:**
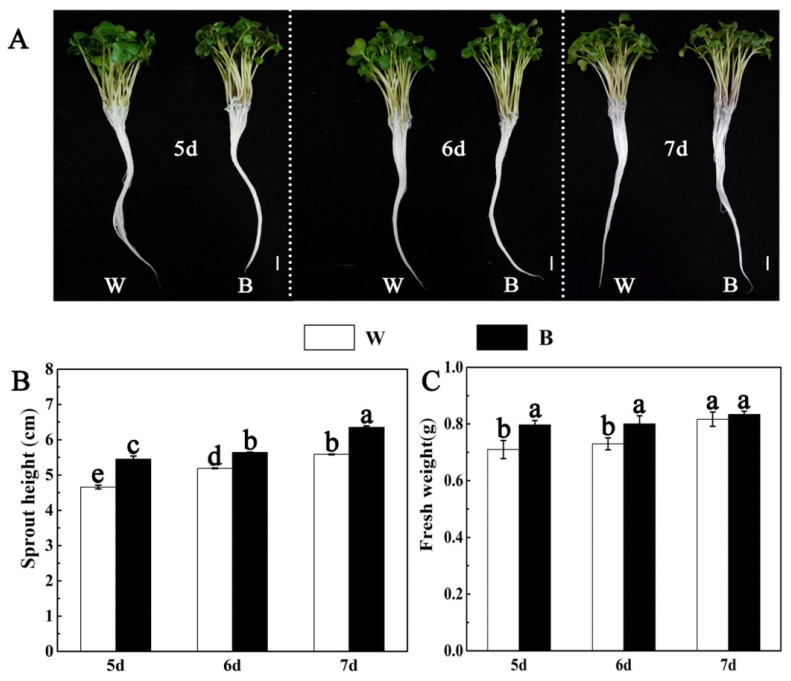
Effect of blue light on growth of broccoli sprouts at different time points. (**A**) Phenotype of 5/6/7 d broccoli sprouts with blue light and white light treatments. Bar = 1 cm. (**B**,**C**) The 5/6/7 d broccoli sprout height (**B**) and fresh weight (**C**) with blue light and white light treatments. Each data point represents the mean of three independent biological replicates (mean ± SE). Different letters indicated statistical differences (*p* < 0.05).

**Figure 2 plants-12-03151-f002:**
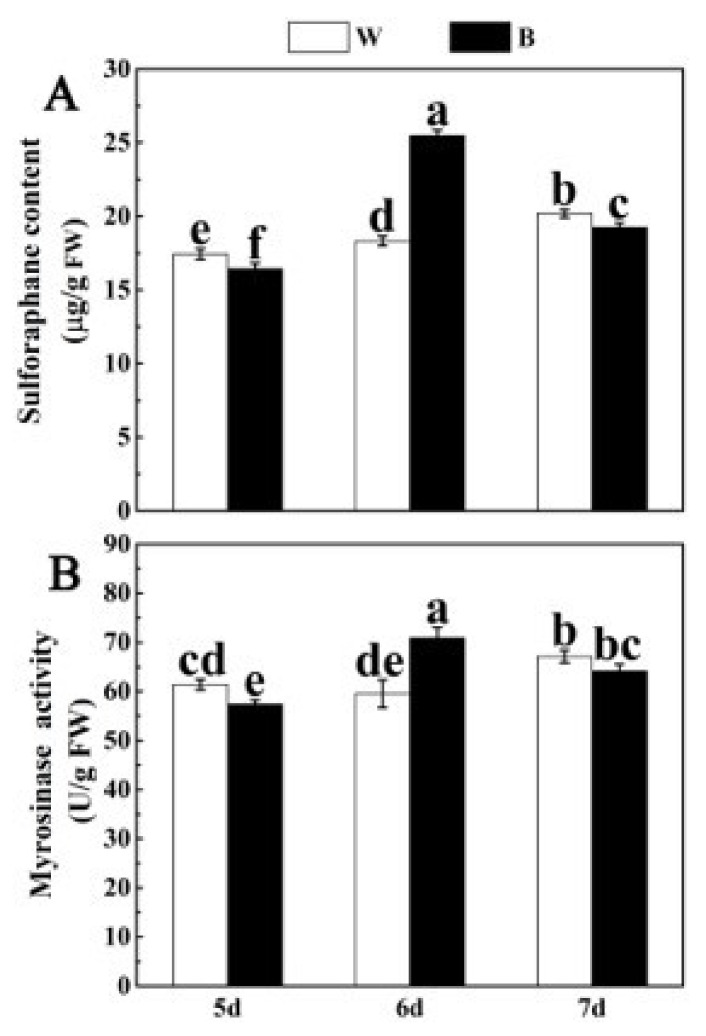
Effect of blue light on SFN content and MYR activity of broccoli sprouts at different time points. (**A**,**B**) The SFN content (**A**) and MYR activity (**B**) of 5/6/7 d broccoli sprouts with blue light treatment. Each data point represents the mean of three independent biological replicates (mean ± SE). Different letters indicated statistical differences (*p* < 0.05).

**Figure 3 plants-12-03151-f003:**
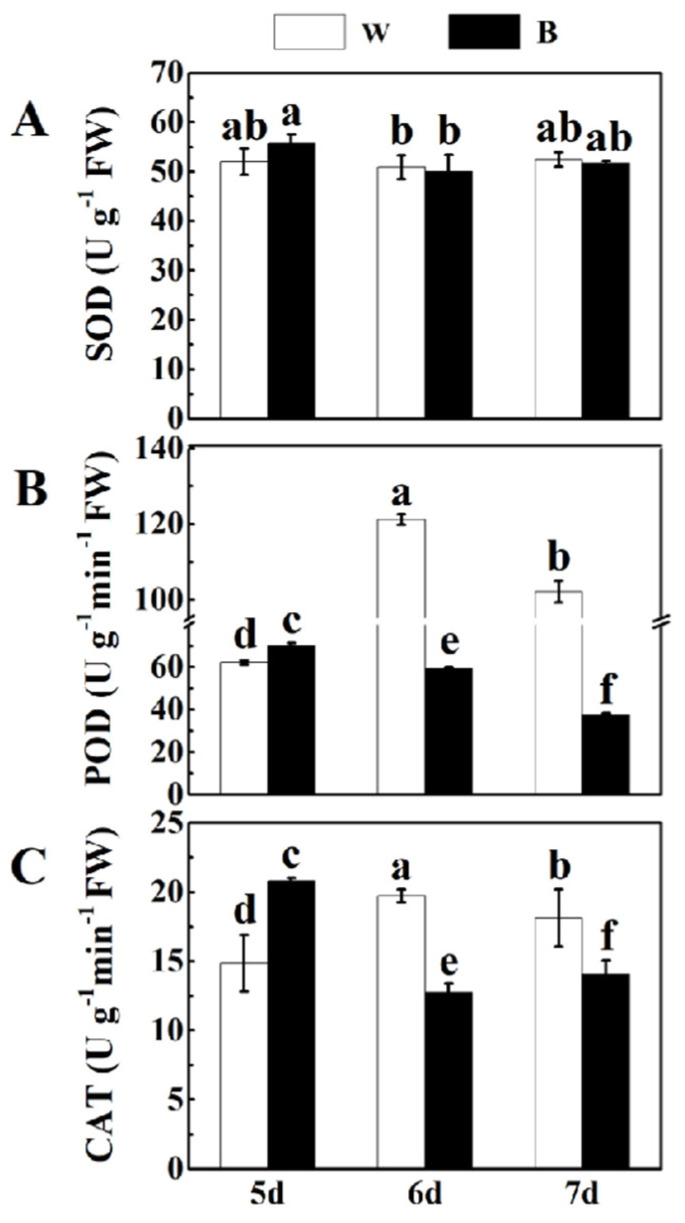
The changes of SOD, CAT and POD activities of 5/6/7 d broccoli sprouts at different time points with blue light and white light treatments. (**A**–**C**) The SOD (**A**), POD (**B**) and CAT (**C**) activities of 5/6/7 d broccoli sprouts with blue light and white light treatments. Each data point represents the mean of three independent biological replicates (mean ± SE). Different letters indicated statistical differences (*p* < 0.05).

**Figure 4 plants-12-03151-f004:**
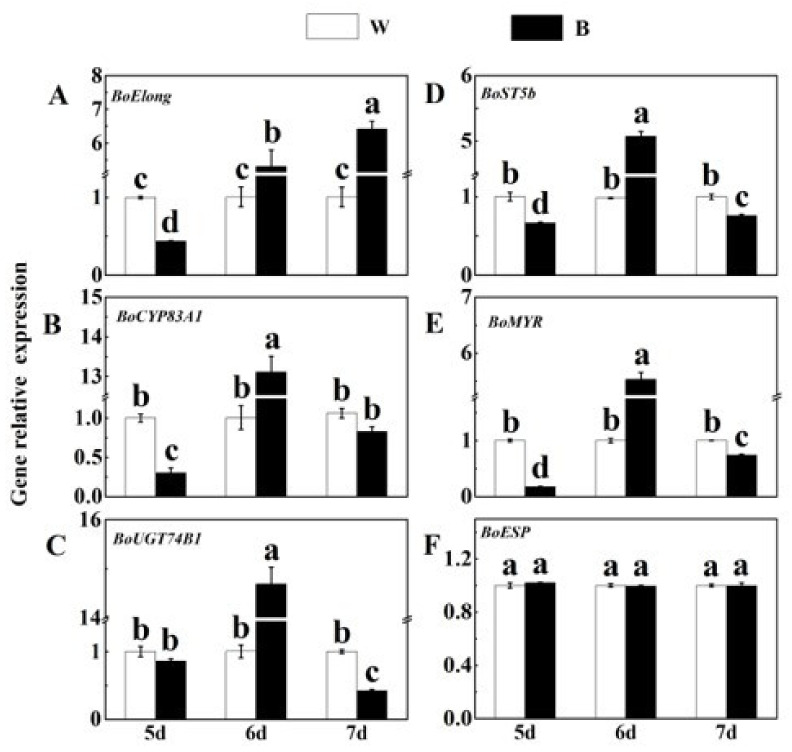
Effects of blue light at different time points on SFN synthesis-related gene expression in broccoli sprouts. (**A**–**F**) Transcript analysis of SFN synthesis-related genes in 5/6/7 d broccoli sprouts with blue light treatment. Each data point represents the mean of three independent biological replicates (mean ± SE). Different letters indicated statistical differences (*p* < 0.05).

**Figure 5 plants-12-03151-f005:**
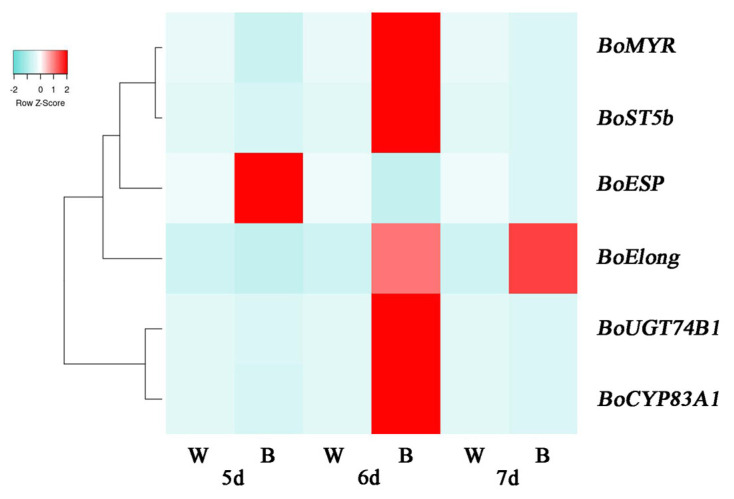
The heat map shows the differences in the expression of SFN synthesis related genes in 5/6/7 d broccoli sprouts with blue light and white light treatments. The values of log 2 [fold change (FC)] were represented using the depth of color, with green representing low expression and red representing high expression. Fold change means the ratio of the gene expression in blue light treatment to it in control.

**Figure 6 plants-12-03151-f006:**
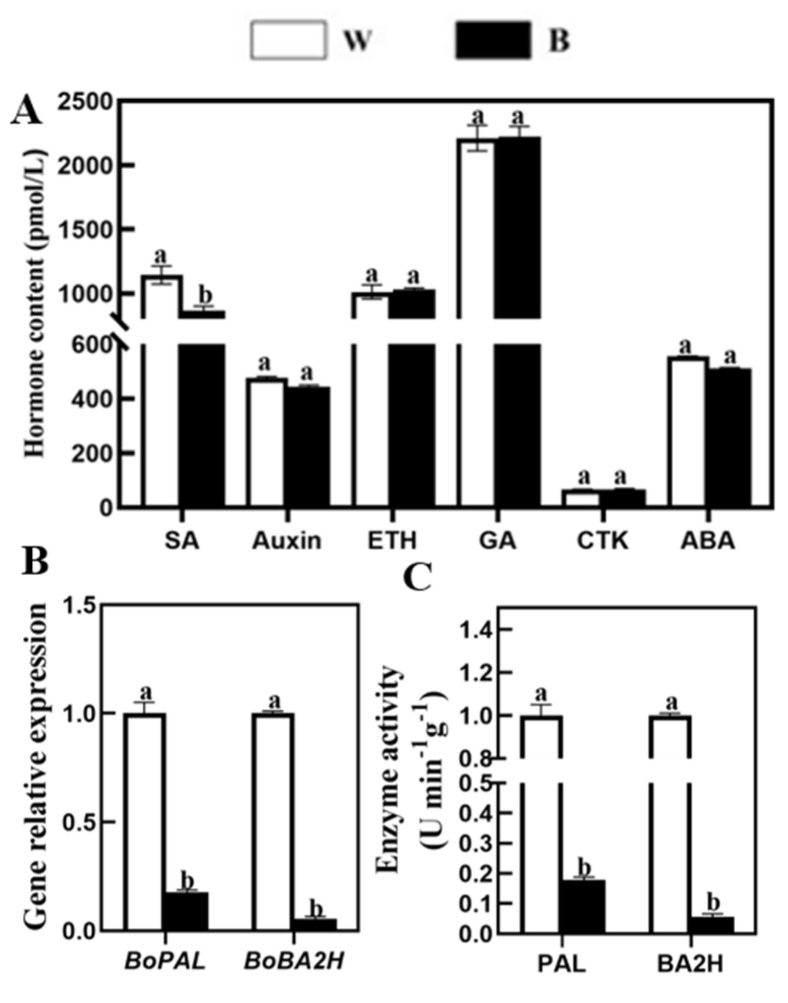
Effect of the content of hormone, key genes and enzymatic activities of the synthetic pathway of SA in broccoli sprouts with blue light and white light treatments. (**A**) The hormone content of 6 d broccoli sprouts with blue light treatment, including SA, Auxin, ETH, GA, CTK and ABA. (**B**,**C**) Transcript analysis of SA synthesis-related genes (**B**) and the activities of key enzymes in SA synthesis (**C**) in 6 d broccoli sprouts with blue light treatment. ETH means ethylene, GA means gibberellin, CTK means cytokinin, and ABA means abscisic acid. Different letters indicated statistical differences (*p* < 0.05).

**Figure 7 plants-12-03151-f007:**
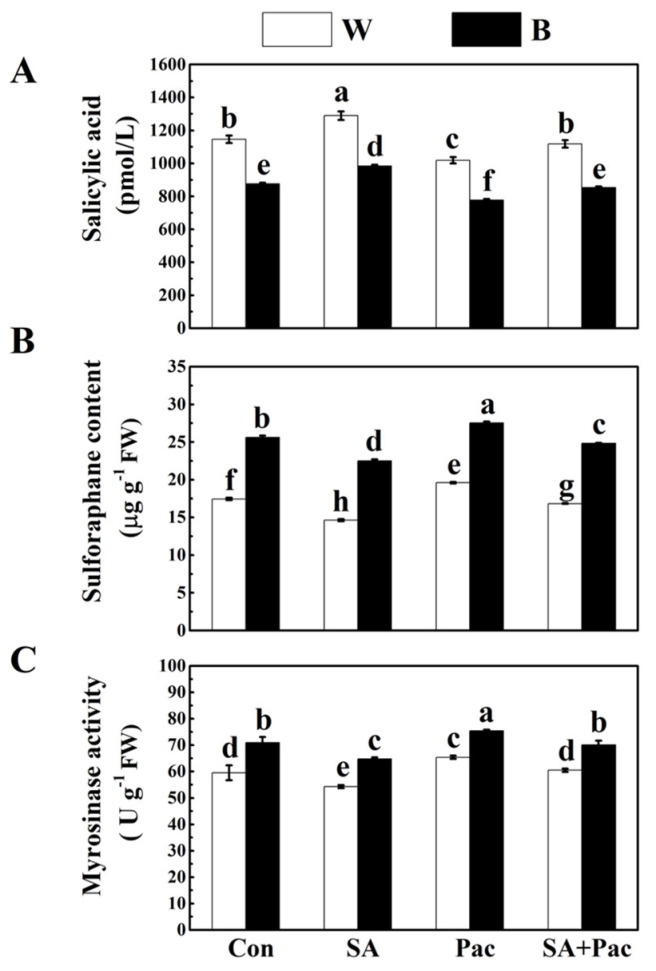
Effect of SA and Pac on SA content, SFN content and MYR activity in broccoli sprouts under blue light and white light treatments. (**A**–**C**) The SA content (**A**), SFN content (**B**) and MYR activity (**C**) of 6 d broccoli sprouts under blue light with SA (50 μM), Pac (100 μM) and their combination (100 μM Pac + 50 μM SA) treatments. Each data point represents the mean of three independent biological replicates (mean ± SE). Different letters indicated statistical differences (*p* < 0.05).

**Figure 8 plants-12-03151-f008:**
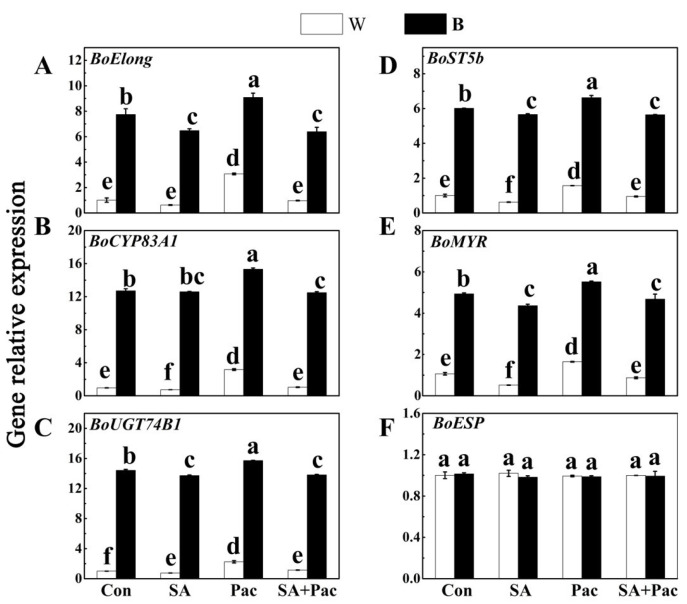
Effect of SA, Pac and their combination on expression of SFN synthesis-related genes in broccoli sprouts under blue light and white light treatments. (**A**–**F**) Transcript analysis of SFN synthesis-related genes in broccoli sprouts with SA (50 μM), Pac (100 μM) and co-treatments (SA 50 μM + Pac 100 μM). Each data point represents the mean of three independent biological replicates (mean ± SE). Different letters indicated statistical differences (*p* < 0.05).

**Figure 9 plants-12-03151-f009:**
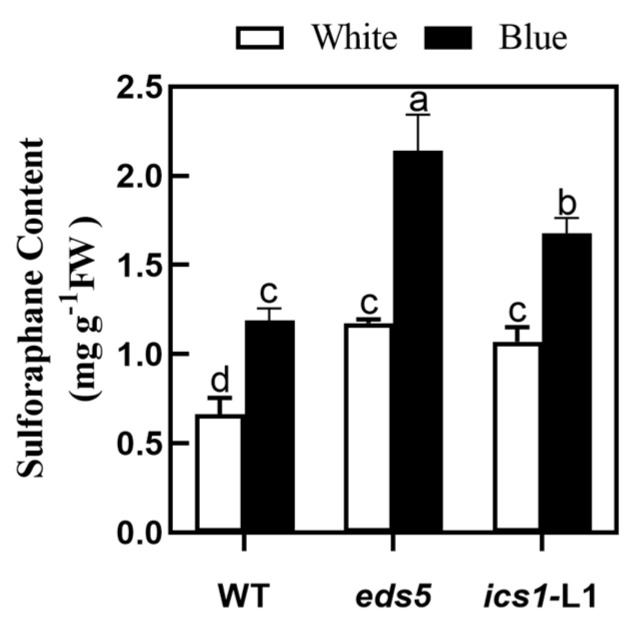
The content of SFN in SA biosynthesis-deficient Arabidopsis mutants *eds5* and *ics1*−L1 under blue light treatment. Each data point represents the mean of three independent biological replicates (mean ± SE). Different letters indicated statistical differences (*p* < 0.05).

**Figure 10 plants-12-03151-f010:**
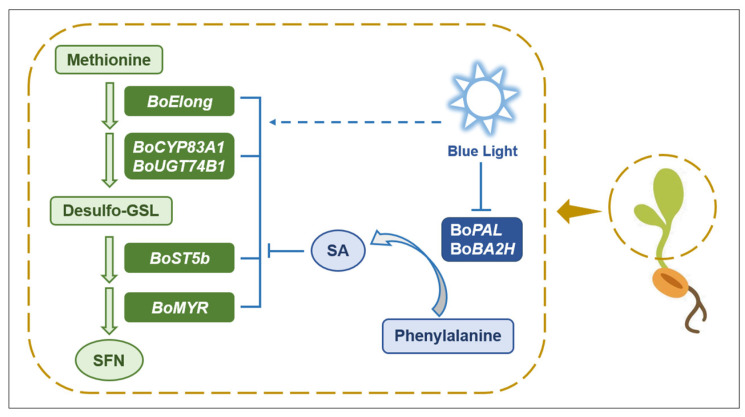
A proposed model of blue light and SA regulating SFN accumulation in broccoli sprouts. On the one hand, blue light can directly promote the expression of SFN synthesis-related genes, including *BoElong*, *BoCYP83A1*, *BoUGT74B1*, *BoST5b* and *BoMYR*, thereby promoting the accumulation of SFN in broccoli sprouts. On the other hand, blue light inhibits the expression of SA synthesis-related genes *BoPAL* and *BoBA2H*. Then, SA biosynthesis was blocked, and the negative regulation of SA on SFN synthesis-related genes was also inhibited, thereby promoting the accumulation of SFN in broccoli sprouts.

**Table 1 plants-12-03151-t001:** Nucleotide sequence of primers in quantitative RT-PCR.

Gene	Primer Sequence (5′ to 3′)	
	Forward	Reverse
*Actin*	CTGTTCCAATCTACGAGGGTTTCT	GCTCGGCTGTGGTGGTGAA
*BoElong*	AAGGTCGTCTGAAAGAGTTGGG	TGATTTCGTTGTCGTTAGTGCC
*BoCYP83A1*	CAAACGCTACAAACTGCC	GTGGGTGAGAACAAGTGG
*BoUGT74B1*	CAAAGACGATAAAGGCTACGGC	TCCCAAAGGAACCAAACGAA
*BoST5b*	CTGATACAGCCTCGCATTG	GAGGGTTTGTGGAATCGTTG
*BoMYR*	ACACGAGATGGCAGAAAG	GACCTCCTTGGTTCACTC
*BoESP*	AAGAGGGAGGACCCGAGGCT	TCCTTTGCTCACTCCACC
*BoPAL*	AGCAGCGGAACAGATGAA	ACTCCCTTTCATCTGTTCC
*BoBA2H*	GCCCTTGTGGTAGCGAAAT	TTGCTCAACACCAGGAAG

## Data Availability

Not applicable.

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
