# Peer review of "Blue Light Enhances Health-Promoting Sulforaphane Accumulation in Broccoli (*Brassica oleracea* var. *italica*) Sprouts through Inhibiting Salicylic Acid Synthesis"

_plants, 2023, doi:10.3390/plants12173151_

Round 1

Reviewer 1 Report

The authors are presenting results on the blue light and glucosinolates in broccoli. Sulforaphane, found in broccoli, is a glucosinolate that is already well-studied for its outstanding anti-cancer effects. Additionally, there are similar papers to this study.

1. recommand changing brassica oleracea L to brassica oleracea var italica.

2. Please change the referecne in main text to mdpi format.

3. Please adjust Figures 1 through 8 to be of the same size and style. They currently appear too scattered and inconsistent.

4. There is some overlap in content between the Introduction and Discussion sections. Please revise the content of the Discussion section to remove redundancy.

5. I kindly request that the Results section be written in a more readable manner rather than simply listing the findings.

While the overall content is quite comprehensible, proofreading and editing of the paper are absolutely necessary to enhance the reader's understanding.

Author Response

Reviewer #1:

The authors are presenting results on the blue light and glucosinolates in broccoli. Sulforaphane, found in broccoli, is a glucosinolate that is already well-studied for its outstanding anti-cancer effects. Additionally, there are similar papers to this study.

  1. recommand changing brassica oleraceaL to brassica oleracea var italica.

Reply:Thanks for your recommandation. We have changed brassica oleracea L to brassica oleracea var italica in the manuscript thoroughly,Line 3, 27, 104.

  1. Please change the referecne in main text to mdpi format.

Reply: We have modified the reference in the manuscript according to mdpi format.

  1. Please adjust Figures 1 through 8 to be of the same size and style. They currently appear too scattered and inconsistent.

Reply:We have modified the Figures1 through 8 in the manuscript.

  1. There is some overlap in content between the Introduction and Discussion sections. Please revise the content of the Discussion section to remove redundancy.

Reply:We have revised the discussion section.

  1. I kindly request that the Results section be written in a more readable manner rather than simply listing the findings.

Reply:We have revised the description of the Results section in the manuscript.

Reviewer 2 Report

The manuscript by Guo et al. describes the positive effect of blue light on the accumulation of SFN, a compound associated with human health benefits. Additionally, the authors correlate this effect with the inhibition of salicylic acid biosynthesis. The experiments were designed properly, the manuscript was well organised and the results, based on combined morphological, biochemical and molecular data, were clear. Therefore, I have the following suggestions to the authors in order to improve the presentation of the results and to make the paper more comprehensive to readers: 

- I believe that it would be better to use the same color in charts for each treatment throughout the manuscript  (e.g. white for "white light" and blue for "blue light").

- Please avoid repeating the names of genes (i.e., in section 3.4 and 3.7) and express the increase/decrease either in -fold times change OR in percentage (choose one of them and use it throughout the manuscript). 

- In section 3.1 the sentence "... the fresh weight was 112.67%, 109.58% and 101.22% of the control, respectively" is not clear (i.e., was it 2-fold times of the control? - probably not), so please use "was increased by ...% compared to control", just like the previous sentence.  

- In section 3.8 (lines 372-375) there are two sentences with exactly the same meaning. Please keep one of them

- In legend of Fig.5 "means" is not necessary. Also, I think that Fig.8 does not show the "Effect of the content of SFN in Acid Biosynthesis Mutants under blue light treatment" as is writen, but "The content of SFN in SA biosynthesis-deficient Arabidopsis mutants". In Suppl. Fig.1 there are no different letters to indicate the statistical differences, as it is written in the legend. Finally, the different letters "indicate" not "indicated" the statistical differences - please correct it in all figure legends

- As you give the primers sequences in the main text (section 2.7), don't write "suppl. table 1" - or move the table to the suppl. material. Further, you should also describe better the reagents and the kits used (e.g. "The total RNA was isolated using the "name of the product" (Invitrogen, Gaithersburg, MD, USA). 

- In section 2.1 you should provide the wavelengths of white light and light. It is important for the reproducibility of the experiment. 

- Finally, I recommend you to have the manuscript read by a native English speaker in order to avoid expressional mistakes (e.g. 261, 273), repeats (e.g. 372) and unnecessary statements (e.g. 213). You did a really nice work and it should be presented correctly to the readers.

The manuscript by Guo et al. describes the positive effect of blue light on the accumulation of SFN, a compound associated with human health benefits. Additionally, the authors correlate this effect with the inhibition of salicylic acid biosynthesis. The experiments were designed properly, the manuscript was well organised and the results, based on combined morphological, biochemical and molecular data, were clear. Therefore, I have the following suggestions to the authors in order to improve the presentation of the results and to make the paper more comprehensive to readers: 

- I believe that it would be better to use the same color in charts for each treatment throughout the manuscript  (e.g. white for "white light" and blue for "blue light").

- Please avoid repeating the names of genes (i.e., in section 3.4 and 3.7) and express the increase/decrease either in -fold times change OR in percentage (choose one of them and use it throughout the manuscript). 

- In section 3.1 the sentence "... the fresh weight was 112.67%, 109.58% and 101.22% of the control, respectively" is not clear (i.e., was it 2-fold times of the control? - probably not), so please use "was increased by ...% compared to control", just like the previous sentence.  

- In section 3.8 (lines 372-375) there are two sentences with exactly the same meaning. Please keep one of them

- In legend of Fig.5 "means" is not necessary. Also, I think that Fig.8 does not show the "Effect of the content of SFN in Acid Biosynthesis Mutants under blue light treatment" as is writen, but "The content of SFN in SA biosynthesis-deficient Arabidopsis mutants". In Suppl. Fig.1 there are no different letters to indicate the statistical differences, as it is written in the legend. Finally, the different letters "indicate" not "indicated" the statistical differences - please correct it in all figure legends

- As you give the primers sequences in the main text (section 2.7), don't write "suppl. table 1" - or move the table to the suppl. material. Further, you should also describe better the reagents and the kits used (e.g. "The total RNA was isolated using the "name of the product" (Invitrogen, Gaithersburg, MD, USA). 

- In section 2.1 you should provide the wavelengths of white light and light. It is important for the reproducibility of the experiment. 

- Finally, I recommend you to have the manuscript read by a native English speaker in order to avoid expressional mistakes (e.g. 261, 273), repeats (e.g. 372) and unnecessary statements (e.g. 213). You did a really nice work and it should be presented correctly to the readers. 

Author Response

Reviewer #2

The manuscript by Guo et al. describes the positive effect of blue light on the accumulation of SFN, a compound associated with human health benefits. Additionally, the authors correlate this effect with the inhibition of salicylic acid biosynthesis. The experiments were designed properly, the manuscript was well organised and the results, based on combined morphological, biochemical and molecular data, were clear. Therefore, I have the following suggestions to the authors in order to improve the presentation of the results and to make the paper more comprehensive to readers:

- I believe that it would be better to use the same color in charts for each treatment throughout the manuscript (e.g. white for "white light" and blue for "blue light").

Reply:Thank you for your advice. We have modified it and used white for "white light" and black for "blue light".

- Please avoid repeating the names of genes (i.e., in section 3.4 and 3.7) and express the increase/decrease either in -fold times change OR in percentage (choose one of them and use it throughout the manuscript). 

Reply:We have modified the contents in section 3.4 and 3.7 for the problem of repeating the names of genes and express the increase/decrease in percentage in the full manuscript.

- In section 3.1 the sentence "... the fresh weight was 112.67%, 109.58% and 101.22% of the control, respectively" is not clear (i.e., was it 2-fold times of the control? - probably not), so please use "was increased by ...% compared to control", just like the previous sentence.  

Reply:We have changed "... the fresh weight was 112.67%, 109.58% and 101.22% of the control, respectively" to " was increased by 12.67%, 9.58% and 1.22% respectively compared to control" in section 3.1,Line 241.

- In section 3.8 (lines 372-375) there are two sentences with exactly the same meaning. Please keep one of them

Reply:We have deleted one of them, Line 418-419.

- In legend of Fig.5 "means" is not necessary. Also, I think that Fig.8 does not show the "Effect of the content of SFN in Acid Biosynthesis Mutants under blue light treatment" as is writen, but "The content of SFN in SA biosynthesis-deficient Arabidopsis mutants". In Suppl. Fig.1 there are no different letters to indicate the statistical differences, as it is written in the legend. Finally, the different letters "indicate" not "indicated" the statistical differences - please correct it in all figure legends

Reply:Thank you for your reminder, we have deleted the "means" of Figure 5.and modified the legend of Fig.8 (Line, 424) and section 3.8 to make them more accurate. Further, we are sorry for the mistake in Suppl. Fig.1 (Line 315-318), we have corrected and checked the full manuscript.

- As you give the primers sequences in the main text (section 2.7), don't write "suppl. table 1" - or move the table to the suppl. material. Further, you should also describe better the reagents and the kits used (e.g. "The total RNA was isolated using the "name of the product" (Invitrogen, Gaithersburg, MD, USA). 

Reply: We have moved table 1 into the main test and described the reagents and kits used, Line 213-216.

- In section 2.1 you should provide the wavelengths of white light and light. It is important for the reproducibility of the experiment. 

Reply: The wavelengths of white light and blue light are 380-750 nm and 460 nm respectively. And we have added it in manuscript, Line 111.

- Finally, I recommend you to have the manuscript read by a native English speaker in order to avoid expressional mistakes (e.g. 261, 273), repeats (e.g. 372) and unnecessary statements (e.g. 213). You did a really nice work and it should be presented correctly to the readers.

Reply:We have corrected these expressional mistakes at Line 261, 273, 372, 213. At the same time, we checked and corrected the expressional mistakes of the full manuscript.

Reviewer 3 Report

Thank you for getting the opportunity to review: “ Blue light enhances health-promoting sulforaphane accumula-2 tion in broccoli (Brassica oleracea L.) sprouts through inhibit-3 ing salicylic acid synthesis” by Youyou Guo, ChunYan Gong, Beier Cao, Tiantian Di, Xinxin Xu, Jingran Dong, Keying Zhao 7a, Kai Gao  and NaNa Su.

The paper provides a description and insight of how blue light can influence the content of sulforaphane in broccoli. This gives an important information to be used to understand the regulation, which can be used to obtain a higher content of these important compounds.

Following is some points for reflection and for updating the paper accordingly, the more detailed comments are included as notes in the commented manuscript.

·       Please visit the results and material and methods, and ensure these are aligned

·       It should be possible to read the legends independent of the text, please revise these so this is possible. As two species are used in this manuscript, it should be noted in the legends when it is results regarding broccoli and Arabidopsis respectively.

·       Please include the rationale behind the choice of hormones tested - and not to include e.g. auxin.

·       It is suggested to inlcude the connection between MYR and SFN in the introduction, for the readers not familiar with GLs.

·       Please reconsider to include the supplementary data, as this figure gives a good visual presentation of the results

Here follows some specific recommendations:

Line 64: this statemetn is in the following - covering both GLs in general and SGN in particular, - one option will be to include this here as well.

Line 85: Here a question, which can be considered to be reflected on - what would be the physiological concentration of SA.

Line 106: as there can be differences between cultivars - the cultivar used in this paper should be stated.

Line 111: details are missing - if the plants are kept in hydroplonics, agar or soil? or are the spouts still kept in the plastic dish?

Line 116: the setup for the experiment for arabidopsis need to be described in more details in the material and methods. Including the mutant phenotype.

Line 120: does this include both Arabidopsis and broccoli?

Line 189: I guess that some words are missing here - The hormone ??? was performed.....

Line 190: It will be good the state which samples are used.

Line 234: please rephrase the sentence starting in line 234

Line 275: in line 275 - please correct effffectively

Line 294: cannot find this part described in the materials and methods. This makes it difficult to relate the results to which concentrations was used, and if these were physiological relevant.

Line 327: please revisit the sentence starting in line 327

Line 377: A describtion of what is phase II enzymes should be included.

Line 461: Please revise the sentence beginning in 461, - especially use the correct terms for " the amount of gene" - the guess is that it is the expression of genes related to ....

The manuscript is easy to follow, however there are a few sentences which need to be revised. See the comments

Author Response

Reviewer #3

The paper provides a description and insight of how blue light can influence the content of sulforaphane in broccoli. This gives an important information to be used to understand the regulation, which can be used to obtain a higher content of these important compounds.

Following is some points for reflection and for updating the paper accordingly, the more detailed comments are included as notes in the commented manuscript.

  • Please visit the results and material and methods, and ensure these are aligned

Reply: We have checked and corrected the results, material and methods in full manuscript.

  • It should be possible to read the legends independent of the text, please revise these so this is possible. As two species are used in this manuscript, it should be noted in the legends when it is results regarding broccoli and Arabidopsis respectively.

Reply:In view of this problem, we have improved and modified all the legends in the manuscript.

  • Please include the rationale behind the choice of hormones tested - and not to include e.g. auxin.

Reply:We measured some common hormone indicators including auxin. In the manuscript, it is mainly described salicylic acid which is most significantly decreased compared with other hormones under blue light treatment, so the auxin measurement index is not reflected in this paper. In order to better understand the article, we have added the measured auxin content to the manuscript. A description of auxin has also been added to the manuscript. Line 329.

  • It is suggested to inlcude the connection between MYR and SFN in the introduction, for the readers not familiar with GLs.

Reply:We have added the content of the connection between MYR and SFN in the introduction, Line 45-49.

  • Please reconsider to include the supplementary data, as this figure gives a good visual presentation of the results

Reply:In order to give the result a good visual presentation, we moved the picture into the main test, Line 315.

Here follows some specific recommendations:

Line 64: this statemetn is in the following - covering both GLs in general and SGN in particular, - one option will be to include this here as well.

Reply:We have revised the literature review cited here, Line 65-66.

Line 85: Here a question, which can be considered to be reflected on - what would be the physiological concentration of SA.

Reply:Thank you for your question. It was found that exogenous application of suitable concentration(2-200 μmol/L)[PerezBalibrea, 2011]of salicylic acid could improve the glucosinolate content and composition of cruciferous sprouts, and high concentration of SA (1-5 mmol/L) [Luo M H, 2009]would reduce the photosynthetic rate and chlorophyll content of plants, thus affecting the growth and development. In addition, through literature search, we found that most of the current studies on the influence of exogenous salicylic acid on the physiological characteristics of broccoli sprouts. The concentration set range was generally 10 μmol/L -300 μmol/L, and the concentration of salicylic acid applied externally in this experiment was 50 μmol/L, which did not significantly affect the growth of broccoli sprouts in the experiment, indicating that the concentration set was reasonable. In this experiment, the SA concentration in broccoli sprouts treated with white light was 1.14*10-6 μmol/L.

  1. PerezBalibrea, Moreno, Diego A etc. Improving the phytochemical composition of broccoli sprouts by elicitation, [J] Food Chemistry,2011
  2. Luo M H, Yuan S, Chen Y E etc. Effects of salicylic acid on the photosystem 2 of barley seedlings under osmotic stress [J]. Biologia Plantarum, 2009

Line 106: as there can be differences between cultivars - the cultivar used in this paper should be stated.

Reply:The variety of broccoli is “Mantuolv”, we have added it in manuscript, Line 104.

Line 111: details are missing - if the plants are kept in hydroplonics, agar or soil? or are the spouts still kept in the plastic dish?

Reply:The spouts still kept in the seeding tray. In order to better understand, we have made further modifications to the method of this part, Line 107-113.

Line 116: the setup for the experiment for arabidopsis need to be described in more details in the material and methods. Including the mutant phenotype.

Reply:We have added details content in the manuscript, Line 116-123. And we have given a supplementary description of the mutant phenotype in this manuscript, Line 415-416.

Line 120: does this include both Arabidopsis and broccoli?

Reply:It only include broccoli sprouts, we have modified it in manuscript, Line 126.

Line 189: I guess that some words are missing here - The hormone ??? was performed.....

Reply:We have supplemented this part in the manuscript, Line 196.

Line 190: It will be good the state which samples are used.

Reply:We have stated which samples are used in manuscript, Line 197.

Line 234: please rephrase the sentence starting in line 234

Reply:We have modified it, Line 253

Line 275: in line 275 - please correct effffectively

Reply:We are sorry for this mistake and have corrected it, Line 304.

Line 294: cannot find this part described in the materials and methods. This makes it difficult to relate the results to which concentrations was used, and if these were physiological relevant.

Reply:We have indicated the concentration used in the method (Line 129-131)

and the legend (Line 368, 404).

Line 327: please revisit the sentence starting in line 327

Reply:We have modified it, Line 377.

Line 377: A describtion of what is phase II enzymes should be included.

Reply: Phase II enzymes are a class of key enzymes in the enzyme system for detoxification of carcinogens, which can detoxify carcinogenic factors. We have added it in manuscript, Line 431-433.

Line 461: Please revise the sentence beginning in 461, - especially use the correct terms for " the amount of gene" - the guess is that it is the expression of genes related to ...

Reply:We have changed" the amount of gene" to" the expression of genes related to SFN synthesis " in manuscript, Line 510.

Round 2

Reviewer 1 Report

The authors have extensively revised sections of the manuscript.

The provided introduction from the research paper discusses sulforaphane (SFN), its properties, synthesis, and the influence of various factors such as light and salicylic acid (SA) on its accumulation in plants, particularly broccoli sprouts.

Introduction

1. The introduction starts with discussing the properties of SFN, then jumps to its synthesis and regulation, the effects of light, salicylic acid, and then concludes with the research objective. While each segment is informative, the flow feels slightly disjointed. A smoother logical progression, starting from the basics of SFN, its importance, factors affecting its accumulation, and finally, the research objective would make the introduction more engaging.

2. The statement "The sulforaphane (SFN), which is identified as the most active health-promoting compound among analogues produced by the glucosinolate (GLs) [1]." is not a complete sentence. The verb is missing.

3. In some places, the paper refers to "GLs", while in other places, it mentions "Gls". Maintaining consistency in terminology or abbreviations is essential for clarity.

 Results and Discussion

1. The discussion regularly cites previous research, making it clear that the authors are well-versed in the topic and have contextualized their work in the broader scientific literature.

2. The study provides molecular insights into how blue light impacts the accumulation of SFN, mentioning specific genes and enzymes. The use of RT-qPCR to track gene expression further solidifies these insights.

3. The research does a commendable job contrasting the effects of blue light and white light, which offers a clearer understanding of blue light's specific impacts.

4. The discussion highlights contrasting findings from other studies, making it evident that the authors are objective and are considering all facets of the topic.

5. While there's an attempt to place the findings in the broader context of existing literature, the section could benefit from more direct statements about the implications of the findings and suggestions for future research.

While the content is largely satisfactory, several sentences would benefit from being rephrased for greater clarity and readability.

Author Response

Introduction

  1. The starts with discussing the properties of SFN, then jumps to its synthesis and regulation, the effects of light, salicylic acid, and then concludes with the research objective. While each segment is informative, the flow feels slightly disjointed. A smoother logical progression, starting from the basics of SFN, its introduction importance, factors affecting its accumulation, and finally, the research objective would make the introduction more engaging.

Reply: We have revised the content of the introduction.

  1. The statement "The sulforaphane (SFN), which is identified as the most active health-promoting compound among analogues produced by the glucosinolate (GLs) [1]." is not a complete sentence. The verb is missing.

Reply: We have corrected it in manuscript, Line 45.

  1. In some places, the paper refers to "GLs", while in other places, it mentions "Gls". Maintaining consistency in terminology or abbreviations is essential for clarity.

Reply: We have changed "Gls" to "GLs" in Line 82, and checked the full manuscript.

Results and Discussion

  1. The discussion regularly cites previous research, making it clear that the authors are well-versed in the topic and have contextualized their work in the broader scientific literature.

  1. The study provides molecular insights into how blue light impacts the accumulation of SFN, mentioning specific genes and enzymes. The use of RT-qPCR to track gene expression further solidifies these insights.

  1. The research does a commendable job contrasting the effects of blue light and white light, which offers a clearer understanding of blue light's specific impacts.

  1. The discussion highlights contrasting findings from other studies, making it evident that the authors are objective and are considering all facets of the topic.

  1. While there's an attempt to place the findings in the broader context of existing literature, the section could benefit from more direct statements about the implications of the findings and suggestions for future research.
